# Alleviation of Severe Skin Insults Following High-Dose Irradiation with Isolated Human Fetal Placental Stromal Cells

**DOI:** 10.3390/ijms232113321

**Published:** 2022-11-01

**Authors:** Boaz Adani, Eli Sapir, Evgenia Volinsky, Astar Lazmi-Hailu, Raphael Gorodetsky

**Affiliations:** 1Biotechnology and Radiobiology Laboratory, Sharett Institute of Oncology, Hadassah-Hebrew University Medical Center, P.O. Box 12000, Jerusalem 91120, Israel; 2Radiotherapy Unit, Sharett Institute of Oncology, Hadassah-Hebrew University Medical Center, P.O. Box 12000, Jerusalem 91120, Israel

**Keywords:** fetal-placental-stromal cells (f-hPSC), cell therapy, skin irradiation, skin histology, wound healing, hair follicles

## Abstract

Skin exposure to high-dose irradiation, as commonly practiced in radiotherapy, affects the different skin layers, causing dry and wet desquamation, hyperkeratosis fibrosis, hard to heal wounds and alopecia and damaged hair follicles. Fetal tissue mesenchymal stromal cells (f-hPSC) were isolated from excised human fetal placental tissue, based on their direct migration from the tissue samples to the tissue dish. The current study follows earlier reports on for the mitigation of acute radiation syndrome following whole body high-dose exposure with remotely injected f-hPSC. Both the head only and a back skin flap of mice were irradiated with 16 &18 Gy, respectively, by 6MeV clinical linear accelerator electron beam. In both locations, the irradiated skin areas developed early and late radiation induced skin damages, including cutaneous fibrosis, lesions, scaring and severe hair follicle loss and reduced hair pigmentation. Injection of 2 × 10^6^ f-hPSC, 3 and 8 weeks following 16 Gy head irradiation, and 1 and 4 weeks following the 18 Gy back skin only irradiation, resulted in significantly faster healing of radiation induced damages, with reduction of wet desquamation as measured by surface moisture level and minor recovery of the skin viscoelasticity. Detailed histological morphometry showed a clear alleviation of radiation induced hyperkeratosis in f-hPSC treated mice, with significant regain of hair follicles density. Following 16 Gy head irradiation, the hair follicles density in the scalp skin was reduced significantly by almost a half relative to the controls. A nearly full recovery of hair density was found in the f-hPSC treated mice. In the 18 Gy irradiated back skin, the hair follicles density dropped in a late stage by ~70% relative to naïve controls. In irradiated f-hPSC treated mice, it was reduced by only ~30% and was significantly higher than the non-treated group. Our results suggest that local injections of xenogeneic f-hPSC could serve as a simple, safe and highly effective non-autologous pro-regenerative treatment for high-dose radiation induced skin insults. We expect that such treatment could also be applied for other irradiated organs.

## 1. Introduction

Upon the introduction of radioactivity and X-ray machines at the beginning of last century the adverse radiation effects to the skin were the first to be visualized, before data on adverse radiation effects to other tissues or organs was collected [1,2,3]. The skin is composed of different tissue layers. The ectodermal tissue consists on a multilayer fast proliferating tightly bound keratinocytes with buds of hair follicles and various skin glands appendages from the dermis. The mesodermal dermis is composed of various connective tissues with slower proliferating cells, in combination with other structures built from various other cell types, including blood vessels cells, glands, neural sensors and resident cells of the immune system. Radiation induced changes can be reflected in the organization of the structures and the width of the different skin layers, as dictated by the proliferation rate of the cells, their death rate, and the regeneration of appendages such as hair follicles [4,5]. Damages to the skin inflicted by high-dose ionizing radiation in different animal models, were found to derive from combined insults to various cell types, with acute expression of damages to the faster proliferating epidermal cells [6]. Though the mouse skin differs from rats or humans, it may serve as an adequate model to examine adverse effects to the different major skin layers [7,8,9,10,11,12].

In-vivo clonogenic assays were applied from the mid 1960’s for evaluations of the survival of progenitors. These tests allowed monitoring the survival rate of epidermal progenitors in relation to the total radiation dose delivered and different dose fractionation in mice [12,13]. Further tests on the same rational were proposed to evaluate the variation of the radiation damages on the whole skin, based on indirect parametric assessment of the physical properties of the skin and healing rate of full depth wounds [7,14]. Other parametric tools for evaluation of late radiation effects to the skin in human subjects were introduced. They included a non-invasive assay of alterations of the mechanical properties of the skin due to late radiation effects, which were tested a few years following radiotherapy [15].

Very limited treatments are available to assist the recovery of radiation damages to skin following high-dose exposure. In an earlier study, before the introduction of the field of stem cells therapies, the injection of expanded isolated autologous newborn’s dermal cells was proposed to accelerate the rate of wound healing in the irradiated mouse skin [16]. Earlier cell therapies with mesenchymal stem cells (MSC) delivery revealed their possible indirect pro-regenerative activity, possibly due to their paracrine effects, without expecting the cells to serve as building blocks of the repaired tissues. The introduction of allogeneic and xenogeneic cell therapies, based on their indirect effect on the regeneration of compromised tissues is a promising approach in the attempts to mitigate radiation effects [17,18].

Bone marrow supportive pluripotent mesenchymal cells (BM-MSCs) were first described by Friedenstein [19]. These and subsequent studies concentrated on the ability of MSCs to differentiate into tissues of mesodermal origin. Though MSCs exhibited in vitro trans-differentiation ability to other mesenchymal tissues, in most cases the effect of their delivery as autologous or allogeneic/xenogeneic MSC did not derive not from their “stemness” with subsequent anticipated contribution to the construction of new autologous tissues [20]. Rather, many studies have focused mostly on their possible indirect effects, by induction of regenerative processes, before their clearance [14,19,21,22].

Accumulated experience suggested that the main positive effect of therapies based on cell implantations it their indirect effects, possibly based on the secretion of a rich relevant secretome by these cells to support the regeneration of the impaired host tissues [21,23,24]. Based on these accumulated insights and data, the term of mesenchymal stem cells was often altered to “therapeutic mesenchymal stromal cells” [25].

The placenta tissues are easily available source of potent mesenchymal stromal cells for various possible allogeneic cell therapies. The profile of the surface markers phenotype of placental mesenchymal cells (PSC) seems to be similar to BM-MSCs. However, they seem to differ in many aspects, such as the rate of their proliferation, their morphology, rate of protein production and their secretome [26].

The tissue site from which the PSC were isolated in our studies, from only the fetal placental tissues Vs the maternal tissues may be associated with their potency to treat severe radiation effects, as previously shown in the mitigation of critical radiation-induced systemic damages [27,28,29,30]. The intramuscular injections of expanded cells, which we isolated from the fetal human placenta tissue (f-hPSC), or commercially produced by Pluristem, were found to be more effective than those isolated from the human maternal tissues for treating acute radiation syndrome (ARS) and other inflammatory conditions in mice [31,32,33,34,35]. The allogeneic/xenogeneic f-hPSC may be less immunogenic than other MSCs, possibly due to their lower expression of different HLA markers, with no expression of HLA-DR [32,36,37,38,39,40,41,42,43,44,45,46,47,48,49,50,51,52,53], which may allow the longer residence of the implanted cells to express their anti-inflammatory and pro-regenerative effect before the cells are slowly eliminated.

Early radiation-induced skin effects may be expressed shortly after radiation exposure with initial damage to the fast-proliferating epidermal cells and hyperkeratosis. Late radiation induced damages to the internal dermal tissues may develop later on, possibly with a long delay [54,55]. Radiation-induced effect on the different skin layers is reflected in a development of hard to heal skin lesions at different time points after exposure, hair loss and possible damages and malfunction of different secretory dermal glands. The f-hPSC therapy is anticipated to modulate these adverse effects, possible by a relevant secretome of the cells [56], though other alternative mechanisms should also be investigated.

The current report shows that subdermal local subdermal injection of xenogeneic f-hPSC at different time points following local high-dose skin irradiation can enhance the regeneration of different target skin tissues. These effects were seen in both preliminary set-up of head only irradiation, and further more detailed experiment on radiation effect on skin in a set-up the high-dose exposure of a large skin flaps with a full shielding of the rest of the body.

## 2. Results

### 2.1. Isolation and Characterization of the f-hPSC

f-hPSC were isolated from both fresh or cryopreserved tissue samples according to our previously reported protocols, as shown in Figure 1 and as previously described in detail [26]. The isolation of the f-hPSC (Figure 1A) is based on their direct migration from tissue fragments dissected from the chorion of male newborn placental tissue to the surface of plastic cell-culture dish (Figure 1B) to form a homogenous pure f-hPSC culture. The migrated f-hPSC are expanded in culture by up to 5–9 passages before injection (Figure 1C). This procedure prevents any contamination with cells from the maternal placenta tissues as well as hematopoietic cells and cells expressing endothelial cell surface markers (Figure 1D).

### 2.2. Head Only Irradiation

An introductive preliminary exploratory experiment was set on f-hPSC treatment following high-dose head irradiation to examine the effect of high-dose head only irradiation with a full protection of the rest of the body. The modulation of radiation damages by subdermal injections of 2 × 10^6^ f-hPSC on days 28 and 46 following irradiation was tested (Figure 2). In this exploratory experimental round, 3 groups of 10 mice were assigned, one arm served as non-irradiated f-hPSC injected control and in the 2 other groups the head only was irradiated with 16 Gy, as shown in Figure 2A with or without f-hPSC injections.

A moderate decrease of body weight gain relative to the non-irradiated naïve control group was recorded in arms of both irradiation only and cell treatment before the f-hPSC injections 3- & 8-weeks post irradiation. During this period, radiation damages were manifested mainly in the thinning and dis-colorization of the head fur and apparent dry skin desquamation (Figure 2A). Unexpectedly, between weeks 8–9 post-irradiation a sharp weight loss of up to ~35% was recorded. The effect was significantly lesser in the experimental f-hPSC treated group (Figure 2A). We assumed that this sharp weight loss at this time point, especially in the non-treated mice, could be not due to other systemic effects but as a response to radiation-induced damages to the exposed continuously growing incisors. This could subsequently induce difficulty to consume the hard bead food. To overcome this issue, the food beads supply was softened by soaking in water and by placing it on the cage floor. This resulted in an immediate sharp weight regain of the mice, so that by the termination of the experiment, approximately 10 weeks following their irradiation, the mice in this group gained back their average weight, relative to naïve controls and the f-hPSC treated mice (Figure 2A). To verify the possible contribution of the radiation induced teeth damage to the transitional reduced weight in the group of irradiated only mice, histology of the incisors’ roots of naïve control and 16 Gy treated mice +/− f-hPSC treatment were taken at the end of the experiment. The sections of the teeth roots confirmed the suspected possible connection of the observed sharp weight loss to severe radiation damage to the incisors of the untreated mice (Figure 2C-2) relative to controls (Figure 2C-1). Of note is the apparent observation of the possible lesser radiation induced damages to these teeth in the f-hPSC treated group (Figure 2C-3).

Examples of the visual records of damages to the irradiated cranial mouse skin on days 46 and 71 following 16 Gy irradiation, are presented in Figure 2B for two time points just before the relevant group of mice were sacrificed. The effects are manifested as areas of dry desquamation, which slowly developed from ~2- to 10-weeks following irradiation, with eventual thinning of the fur associated with visually observed alopecia in the irradiated field and very significant hair loss with lost hair pigmentation. The f-hPSC treatment in this preliminary experiment was associated with visually apparent protection from these radiation effects.

Upon termination of the experiment on day 71, skin samples from the center of the irradiated area were harvested, fixed in their original flat configuration, embedded and sectioned in parallel to the skin surface. The sections were taken in the deeper stratum basal epidermis on the boarder of the dermal tissue, in parallel to the skin surface. This enabled to cut through the hair follicle roots and perform a morphometric analysis in the skin sections, with quantitation of hair follicles density per mm^2^ of sectioned tissue (Figure 2D).

The histological quantitative assay of hair follicles density at the termination of the experiment, 71 days following irradiation, showed a drop of the density of hair follicle by more than 50% in the irradiated skin (Figure 2D). The quantitative summary of the results of the density of the hair follicles in mice that were treated on day 46 post-irradiation by subdermal f-hPSC injection is shown in Figure 2E. A significantly lower hair follicles count was recorded in the irradiated untreated mice, relative to f-hPSC treated mice, whose hair follicle counts were not significantly different from the non-irradiated f-hPSC treated controls (Figure 2E).

### 2.3. Effects of High-Dose Irradiation Restricted to a Large Skin Flap with Full Protection of the Rest of the Body

Based on the results of the preliminary experiment on the head only irradiation a more detailed set of experiments was that focuses on the effect of exposure of a back skin flap to higher dose of 18 Gy with full protection all organs in the rest of the body was instigated. A higher dose was delivered to the skin flap and the f-hPSC were injected within shorter time intervals from the skin irradiation, as shown in Figure 3A,B.

In this set-up, once all possible radiation effects on other organs were eliminated, the skin irradiated mice gained weight similarly to both non-irradiated control groups until the termination of the experiment on day 52 (Figure 3B). In the set-up of this experiment the subdermal f-hPSC injections under the irradiated skin area was performed earlier, one week and four weeks following irradiation, before the apparent onset of late skin adverse effects.

The earlier time point for the final termination of this experiment allowed a detailed monitoring of the histological changes at the peak of the late radiation damages. It also enabled the evaluation of the possible effect of the treatment in adequate timing, in case such detectable differences may fade in longer follow-up. Detailed photographs of the back skin of all mice were taken on a weekly follow-up, as shown for representative mice from each arm of the experiment (Figure 3C). As evident from these photographs, the irradiated skin with no f-hPSC treatment developed more severe lesions with wet and dry desquamation from ~4 weeks following irradiation onwards, leading to open ulcerative wounds on top of apparent alopecia. Some of these conditions persisted up to the termination of the follow-up. By contrast, the hair loss and alterations in the skin of the f-hPSC treated mice 28 days following irradiation were apparently milder, with only minor lesions. Almost a full recovery from many of these lesions was observed by day 53 following irradiations (Figure 3C).

Approximately half of the mice in this detailed experiment were sacrificed for histology on an earlier time point on day 28 following irradiation, and the rest on day 53, before further progression the skin healing occurred, which could reduce the difference between the various groups tested. External observation of the mice showed that the hair of the f-hPSC treated mice seemed to recover better with apparent regeneration of hair density. The loss of the hair pigmentation in the irradiated skin, as seen mostly in the f-hPSC treated mice, where the non-pigmented hair dominated in the recovered fur, hints for the higher radio-sensitivity and/or slower recovery of the melanocytes within the regenerated hair follicles, (Figure 3C). The grading of all these parameters in visual double blind experts’ evaluation was averaged and summarized for each group tested, as shown in Figure 3D. The two control groups with only cell injection were found to express minimal alterations, within the low-rank range of 1–2 along the follow-up. These minor skin irritations at the beginning of the experiment may have derived from the skin shaving and sham irradiation of the non-irradiated controls at the onset of the experiment.

The quantitation of hair follicle density in the histological samples were done on histological sections, taken in parallel to the skin surface in deeper stratum basal epidermis layer as shown in Figure 2D, similarly to the procedure described for the preliminary experiment. H&E stained Skin sections of three representative mice in each of the groups tested are shown in Figure 4A. The summary of the analyses of the hair follicles density per mm^2^ of the dermal tissue area per for naïve controls and irradiated skin with/without f-hPSC treatment is presented in Figure 4B. In spite of the high dose delivered and the visually apparent changes in the mice skin and fur, as evident on day 28 following irradiation, the radiation effect as expressed by hair follicle density was still mild and not significant. Nevertheless, on day 53, at the termination of the experiment, a highly significant drop in hair follicles density was observed in the skin of the non-treated irradiated mice, while only moderate non-significant decrease in hair follicles density was monitored f-hPSC treated mice (Figure 4B).

A follow-up of skin rigidity by non-invasive VESA measurements was performed during the experiment on a weekly basis for experiment 2, as presented in Figure 4C,D. A significant reduction of the rigidity of the irradiated skin was recorded 4–5 weeks from irradiation (Figure 4C,D). Since the VESA allows detecting skin anisotropy it could show that changes in the skin rigidity were more pronounced and significant in the lateral/medial direction (Figure 4D). The parallel non-invasive long follow-up of surface skin moisture level by CK-Corneometer showed a highly significant elevation of the skin moisture in the same time interval, which returned to normal values from ~40 days following the skin irradiation, hinting for possible parallel wet inflammatory reaction at that time interval (Figure 4E).

The measurement of the average thickness of the different skin layers was done based of the H&E-stained skin cross-sections of all the mice with internal area calibration based on the photographs. Representative histological cross-sections on days 28- and 53-post-irradiation are given in Figure 5A. The summary of the average thickness is presented for the epidermis (Figure 5B-1), dermis (Figure 5B-2), and adipose layer (Figure 5B-3). The most significant effect of the high-dose irradiation was the highly significant increased thickness of the skin epidermal layer in both irradiated groups, as seen on day 28 following irradiation, with no effect of the f-hPSC treatment. These early radiation effects faded somehow by the termination the experiment on day 53 following irradiation, possibly due to some age-related small increase in the epidermal thickness in all the arms tested. As to the dermis, a small but still significant transient increase in its thickness was seen on day 28 after irradiation, possibly due to a mild dermal edema, with no apparent effect of the f-hPSC treatment. No difference from non-irradiated naïve controls was observed for the dermis by day 53. As to the adipose layer, the data observed were variable which could not allow to point on significant changes. This includes a non-significant effect in the irradiated mice 28 days following skin irradiation, which faded at the end of the follow-up on day 53.

Evaluation of fibrotic collagen deposition in the skin were done on Mason’s trichrome stained histological sections 28 and 53-days following irradiation, as shown in Figure 5C. Morphometric evaluation of collagen deposition was done by counting the blue stained pixels in the Mason’s trichrome stained skin sections per the exact net area of the tissue section, as translated from pixels to mm^2^ (Figure 5B). A certain non-significant increase in blue staining, hinting for minor elevation of fibrosis was seen in the two groups of irradiated mice by day 28. These changes were not apparent at the termination of the follow-up on day 53. These data indicate that a most apparent effect of f-hPSC treatment is the alleviation of skin damages following high-dose irradiation. The most significant contribution of the treatment seemed to be the preservation of the hair follicles density, which were most affected by the high doses of 16–18 Gy irradiation.

## 3. Discussion

Efficient therapies for the protection of severe delayed effect of high-dose irradiation are scarce. As shown in our previous reports, the selected expanded fetal derived placental stromal cells injected even a few weeks, following skin irradiation can serve as an efficient indirect cell therapy to expedite the regeneration of heavily irradiated tissues.

We could demonstrate in the current study that local subdermal f-hPSC injections at different time points following irradiation had a major effect on the skin recovery from radiation damages. The f-hPSC injection to pre-irradiated skin, when the damages were already apparent, induced significant pro-regenerative effects, predominantly in preventing the depletion of the sensitive skin hair follicles. The effect of the f-hPSC injections was significant both in whole head irradiation with a lower dose, or in a local higher dose irradiation of only a back skin flap. The data suggests that the optimized timing of the f-hPSC injections after irradiation may deserve further investigation. The main measurable effect of the f-hPSC injections was their pro-regenerative activity on the radiosensitive hair follicles. Other parameters tested, which included histological morphology, showed temporary alteration in the thickness of certain skin layers, such as the epidermis with time related reversal of the effect, which faded with time upon the recovery of the skin from the radiation insults.

The main observed effect of the f-hPSC treatment was the highly significant induction of the regeneration of hair follicles in the irradiated skin. This opens a new promising direction to study the development possible treatment by f-hPSC injections for the prevention of common non-radiation induced alopecia to which so-far no commonly accepted practical solutions are available.

Of interest was the unexpected finding in the preliminary experimental set-up, with whole cranial irradiation, of a severe reversible few weeks weight loss at the late stage of the follow-up. This effect was significantly lower in the experimental arm of subdermal f-hPSC injection. We assumed that this effect might have derived due to a significant radiation induced damage to the open roots of the mice incisors, thereby affecting their food intake. The effect seemed to be much lesser in the f-hPSC injected mice. Though the f-hPSC were injected under the skin scalp, the apparently indirect effect of the f-hPSC treatment possibly reached the affected teeth roots, thereby reducing significantly the long-term radiation damages to the mice incisors, as further verified by histological sections of the mandible at the end of the experiment. Based on these preliminary observations further dedicated studies should explore the possibility of reducing dental damages to heavily irradiated jaws by adjacent subcutaneous f-hPSC injections in high-dose irradiation. This supports previous studies showing that f-hPSC could also serve as a therapy for inflammatory and degenerative insults in other conditions, such as inflammatory damage to the brain [35].

In summary, the current report supports the feasibility of using apparently complication-free f-hPSC treatments as a simple indirect pro-regenerative treatment for various radiation-induced tissue insults that could be possibly applied for other disorders associated with severe degenerative processes.

## 4. Materials and Methods

### 4.1. Isolation, Expansion and Characterization of the f-hPSC

The methodology for f-hPSC isolation was previously described in detail [26]. The use of the disposed placentae was approved by Institutional Helsinki Committee with further approval of the Israeli Ministry of Health (no.: HMO-14-0361-920140112) with the donating parents’ informed consent. By our protocols the f-hPSC could be isolated from both fresh or cryopreserved tissue samples as previously reported [26]. Briefly, the fresh donated placentae were collected following elective Cesarean operation of health child. Only placentas of male offspring were used to enabled to verify the fetal source of the isolated hPSC. The X and Y chromosomes were stained by FISH to verify their male fetal origin. The placentae were handled in sterile conditions until their immediate processing. Tissue samples from the male chorion tissue layer were chopped to small fragments (~1–3 mm) which were lightly digested with trypsin for 10–20 min in 37 °C CO_2_ incubator. Then, the chopped tissue fragments were immersed in culture medium with diluted fibrinogen (~1 mg/mL). The tissue fragments were then dispersed on the surface of a large cell culture dish and the residual liquids were carefully evacuated. The moist fibrinogen coated fragments were immobilized on the plastic dish by light spray of 100 U/mL thrombin. Following ~15 min incubation for fibrin mediated immobilization of the moist coated fragments to the plastic surface, f-hPSC culture medium (DMEM low L-glucose with 1% L-glutamine, 1% pen-strep and 10% FCS of Gibco (USA), all supplied by BI-Industries-Sartorius, Beit-Haemek, Israel) was carefully added and the plates were placed in 37 °C in a humidified 6% CO_2_ tissue culture incubator. A slow cell migration from the tissue fragments fibrin immobilized to the plastic cell culture took place beginning a few days later, as previously described and shown now schematically in Figure 1.

Isolated f-hPSC expansion was performed in tissue culture flasks with the above f-hPSC medium, typically with at least one medium exchange per week. Cells from passage ~4–8 were used for implantation. To rule out mycoplasma contamination in the expanded f-hPSC, the cultured cells were periodically tested during their expansion with a dedicated qPCR Detection kit (Sigma-Aldrich, Jerusalem, Israel) and/or with MycoBlue colorimetric Mycoplasma Detector test (Vazyme biotech, Dusseldorf, Germany).

The pure fetal origin of cell batches, isolated from different excised placental tissue samples, was verified by FISH analyses of the centromeres of their X and Y chromosomes. Cells on passage 2–3 were plated on a cover slip for the FISH analysis (Vysis, Abbott Molecular, Abbott Park, IL, USA) and the X/Y chromosomes analyses were performed at the Clinical Genetics Department Labs of Hadassah Medical Center [26].

### 4.2. Mice

C3H/HeNHsd female mice, ~7 weeks old, were purchased from Harlan/Envigo-RMS Israel Ltd. and acclimated for at least 5 days before the initiation of the experiments. The choice of female mice for this model is due to their relatively slower growth rate and their less aggressive behavior in prolonged follow-up. The mice were irradiated and kept during the whole experiment and follow-up in specific pathogen free (SPF) conditions at Hadassah Hebrew University animal colony (ISO 9001:200, Ethical Animal Welfare Certificates #GB06/68708 and Institutional Animal Welfare Committee of the Hebrew University of Jerusalem MD-12-13296-4 and MD-16-14727-4). At the onset of the experiment, the mice weight was in the range of 18–21 g. To enable the follow-up of weight change relative to their initial weight, the mice weight was normalized for each individual mouse to its initial weight and the % changes are presented.

### 4.3. Irradiation

Irradiation was performed with 6MeV homogenous dose electron beam of the field of interest, delivered by clinical LINAC device with SSD of 100 and a build-up of 5 mm soft gel polymer placed over the exposed area of the jigs, as shown in Figure 2A and Figure 3B. Special 2 types of irradiation jigs were constructed to either expose the head or exposed only a large back skin flap with full protection the rest of the body. The evaluation of the accurate dose delivered to the exposed mice was calculated by radiotherapy physicists of the Radiotherapy unit at Hadassah Hospital. The calibration of the penetrated dose distribution through the shield was performed by the operation specialized physicists with the LINAC software, based on measured isodose outline as calculated by the linear software. Additional verification dosimetry was done with radiation mini-probes, performed on a mockup set-up with a mouse cadaver loaded in the irradiation jig with the led shielding of the non-exposed areas as performed in the experiments. The small background doses measured are minimal and close to threshold and they are given just as a reference.

Before skin irradiation in the 1st phase of the head irradiation experiment, the upper head skin was shaved. Before the back skin irradiation. In irradiated mice, the whole head was exposed to a dose of 16 Gy with protection of the rest of the body, as shown in Figure 2A. In the large back skin flap irradiation a flap area of >1.5 × 2 cm was shaved and then exposed to 18Gy, with protection of the rest of the body. The protective 3 mm lead shield in the jigs allowed >95–97% dose reduction of the unexposed areas (estimated exposure of the protected mice <0.8 Gy). The mice were anaesthetized and the skin of the back was shaved, a flap of ~1.5 cm × ~2 cm was carefully pulled through a slit in the lead shield and secured non-traumatically on the upper part of the jig lead shield by a surgical adhesive plaster, as illustrated in Figure 3A. In the back skin flap a higher dose of 18 Gy was delivered with exposure of the shielded organs to anticipated calculated scattered background dose of less than 1 Gy.

### 4.4. Cells Injection

Harvested cells were collected in buffered Plasmalyte injection solution (Baxter). The cells were diluted to 2 × 10^7^ cells/mL in Plasmalyte. In all the cell treatments 100 µL with 2 × 10^6^ f-hPSC were slowly injected subcutaneously to the treated mice. The control untreated mice were injected with Plasmalyte alone.

### 4.5. Parameters Tested to Follow Up the Irradiated Skin Condition

The preliminary experiment of head irradiation the follow-up was limited to the evaluation of relative weight gain/loss and records of detailed photography analyses of the irradiated head skin with quantitative histology of the skin layers and hair follicle density.

Based on the feedback from the initial preliminary scalp irradiation the 2nd part of the study focused on the effect of skin-only irradiation with the full protection of the rest on the body, for evaluating of the radiation induced skin only damage without possible influence of radiation insults to other organs. The irradiated skin surface along the follow-up was documented weekly with detailed serial photographs in standardized condition of all the mice. Additional parameters that were tested in parallel included the external skin surface (stratum corneum) hydration measurements by Corneometer^®^ CM 820, whose operation is based on external electrical conductance of the skin (Courage-Khazaka (CK) Electronic, Köln, Germany).

The evaluation of changes in skin viscoelasticity was done by the Viscoelasticity Skin Analyzer (VESA), a device previously developed by our group (Gorodetsky et al., US patent 4947851A), for the non-invasive measurement of the mechanical properties and viscoelasticity of the skin, based on the speed of propagation of directional traveling acoustic wave on the its surface [57]. The VESA allows the directional evaluation of the viscoelasticity in proximal and distal (lateral/medial) directions to monitor also possible skin anisotropy (Figure 4C,D), as previously applied in studies with non-invasive follow-up of late radiation skin damages in a human study [15].

### 4.6. Professional Expert Review of Histology to Quantify Skin Damages Grading of Skin Condition

Grading of the general skin condition was done by averaging double-blind evaluation by two experts of two rounds each (by R.G. and E.S), the weekly taken detailed photographs of all the mice. The blinded scoring for damages in the irradiated skin was performed by two independent experts, as shown in Figure 3D. The scoring is based on a scale that was designed for this study. The scoring parameters were as follows: minor inflammation and redness, with no other apparent skin lesions or damages was graded as 1 to 2, dependent on the severity of this condition. Apparent dry desquamation was graded 3. Minor scattered superficial lesions on the skin with no open wounds was graded 4. Wounded skin with wet desquamation was graded 5 and damaged skin with epidermal loss and significant necrotic areas was scored as 6.

### 4.7. Samples Preparation for Histology

For skin histology, at the end of the experiment the boundaries of the irradiated skin flap area of interest were excised a its pre-marked boundaries. The skin samples were then placed in their original planar orientation on a flat hard filter paper for 24–48 h and fixed flat in 4% formaldehyde solution. Then, the samples were transferred to 70% ethanol until their embedding. The skin samples were placed horizontally in the paraffin block, parallel to the surface of the sections. Sections were taken through the upper dermis, below in the inner epidermal layer, to enable the accurate morphometric analyses of the hair follicle roots density. In parallel, skin strip samples were also embedded for cross-sections. Sections were stained with H&E or Masson trichrome by previously described protocols [33,34].

Based on the finding of possible radiation-induced damage to the teeth along the follow-up in the irradiated heads, histology of the teeth roots was also added. For preparation of teeth and adjacent skull bone samples for histology, following the 24–48 h of 4% formaldehyde fixation, the samples were demineralized extensively for 30 days in concentrated buffered EDTA solution, with twice weekly solution exchanges, as previously described [34,58]. This allowed minimal detachment and tearing of the structures from adjacent soft tissues during sectioning.

### 4.8. Histological Morphometry

All tissue-stained sections were viewed, scanned and photographed with computerized Nikon light/fluorescence microscope, equipped with a camera with an internal calibration adjusted the relevant magnification. Serial consecutive photographs were taken for long tissue sections. These photographs were carefully “stitched” together with Photoshop image processor (ver. 7 or higher) to allow for the presentation of the full view of long sections. For presentation of areas of interest with transformation from pixels to mm^2^, a standard cell-counting glass chamber with accurately marked 1 × 1 mm scale was photographed in the same magnification (typically ×10) of the slides. The number of pixels within the area of 1 × 1 mm as presented by the image processor program allowed a calculation of the tissues section area size in mm^2^, as well as areas of different layers of interest within the section.

For the evaluation of the area of stained tissue of interest in the histological slides, the number of pixels of the color of interest within the marked stained tissue section were evaluated relative to the total tissue pixels. For counting the number of sectioned structures of interest, such as hair follicle density, they are presented relative to the net total issue area on the slide in which they were counted.

### 4.9. Statistical Analyses and Data Presentation

All error bars presented represent the standard error of the mean (which is also influenced by the size of the sampled data). Statistical analyses for the comparison between the tested groups were done by multi-parametric Student’s *t*-tests, assuming equal variances, with one-way analysis of variance and post-hoc corrections for multiple group comparison, when applicable in MS Excel databases and statistical analysis package. The relevant built in applications were used for data analyses and evaluation of the statistical significance of compared data, as also elaborated in relevant publications [59].

## Figures and Tables

**Figure 1 ijms-23-13321-f001:**
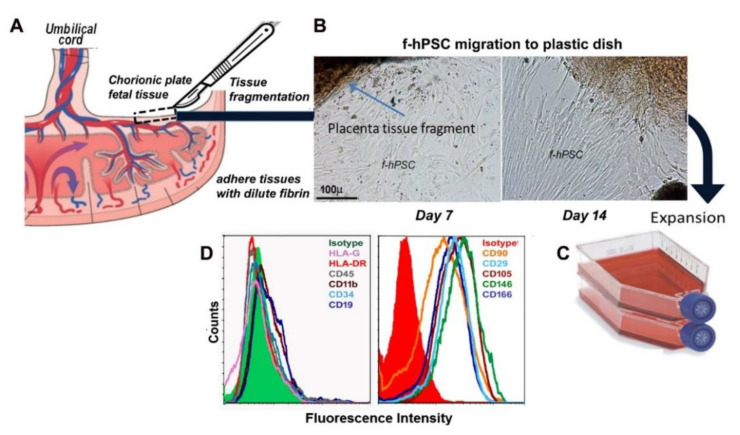
(**A**) A scheme of the placenta structure with adjacent tissues from two separate individuals, the fetus and the mother, each with separate network of blood vessels. f-hPSC are isolated from tissue sample carefully dissected out from the fetal placental tissue only. (**B**) Small tissue samples are dissected from the chorionic plate are chopped into small tissue fragments which are adhered by dilute fibrin glue to the plastic surface, allowing the direct slow f-hPSC migration from the tissue fragments to the culture dish to reach a homogenous population of f-hPSC. (**C**) The isolated f-hPSC are spontaneously migrating out through dilute fibrin glue to culture flasks for further expansion for 5–9 passages, before their harvest and injection suspended in Plasmalite isotonic buffered solution. The profile of the surface markers phenotype of the cells is presented in (**D**).

**Figure 2 ijms-23-13321-f002:**
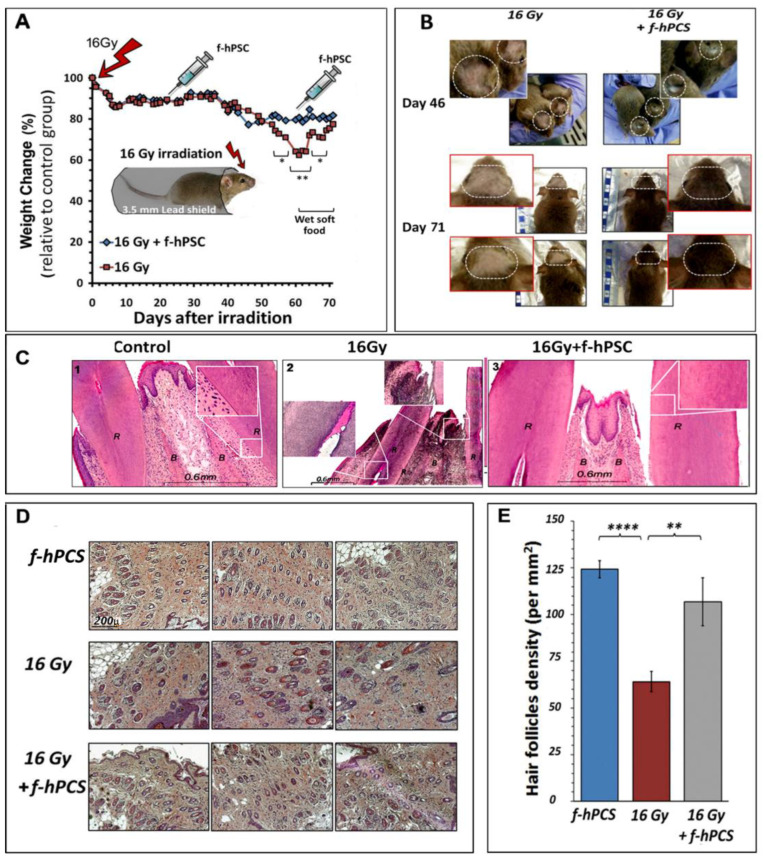
The set up and results of the preliminary model of head only irradiation is shown in (**A**). The whole body was fully protected and only the skull from the ears forward was exposed to the electron beam irradiation. The follow-up of weight change normalized to the values recorded from naïve non-irradiated controls is presented. (**B**) The skin of the front cranial 71 days after irradiation. (**C**) Representative histology of the roots of the incisors in the different groups tested with section through the cranial bone tissue (marked as B) and the open roots of the incisors (marked as R) are presented for the control (1), irradiated (2) and irradiated and f-hPSC treated (3) (**D**). Typical H&E histological sections parallel to the skin surface in the upper dermis for evaluation of hair follicles density per mm^2^ in different groups tested on day 71. All photographs were taken at the same magnification (**E**). A summary of the histological morphometry analysis of the hair follicle density of each experimental group tested (** *p* < 0.01, **** *p* < 0.001).

**Figure 3 ijms-23-13321-f003:**
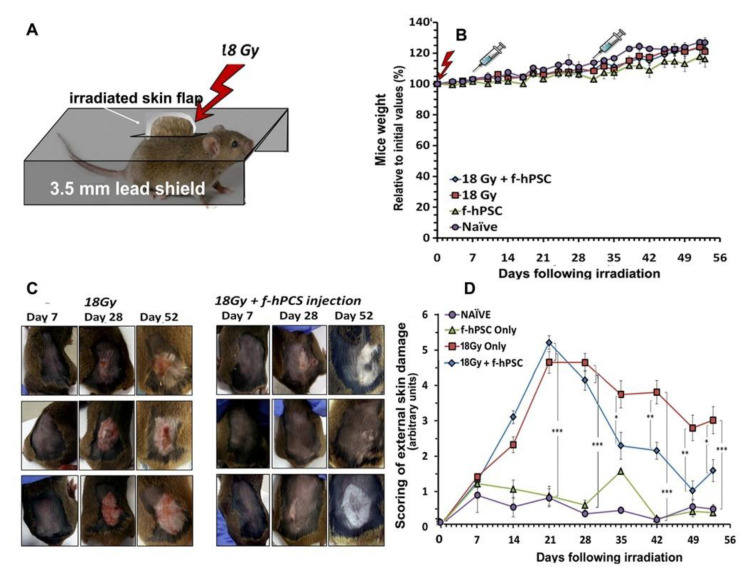
The setup of the experiment on the exposure of skin only to 18 Gy irradiation (**A**). The mice were fully shielded and a marked boundary skin flap is pulled through a slit in the lead jig, which shields the rest of the body. (**B**) The follow-up of the general condition of the mice as reflected in their weight gain. The f-hPSC treatments were delivered as local subdermal injections under the irradiated skin, with shorter time-intervals from irradiation, as compared to the preliminary experiment with head only irradiation, as shown in (**B**). The f-hPSC injection time points are marked on this graph. No significant difference was evident in weight gain between the experimental groups and naïve untreated and f-hPSC treated controls (**C**). Presentation of the visually recorded condition of skin flaps in 3 representative mice for 18 Gy +/− f-hPSC treatment showing the development of lesions along the follow-up in the irradiated skin and their better recovery in the f-hPSC treated mice, with regain of hair growth. A loss of hair pigmentation in certain degree in the recovered fur was apparent in most mice. The average blinded expert scoring of skin condition based on the photographed records of all mice in each of the different groups tested along the follow-up is shown in (**D**) (* *p* < 0.05, ** *p* < 0.01, *** *p* < 0.005).

**Figure 4 ijms-23-13321-f004:**
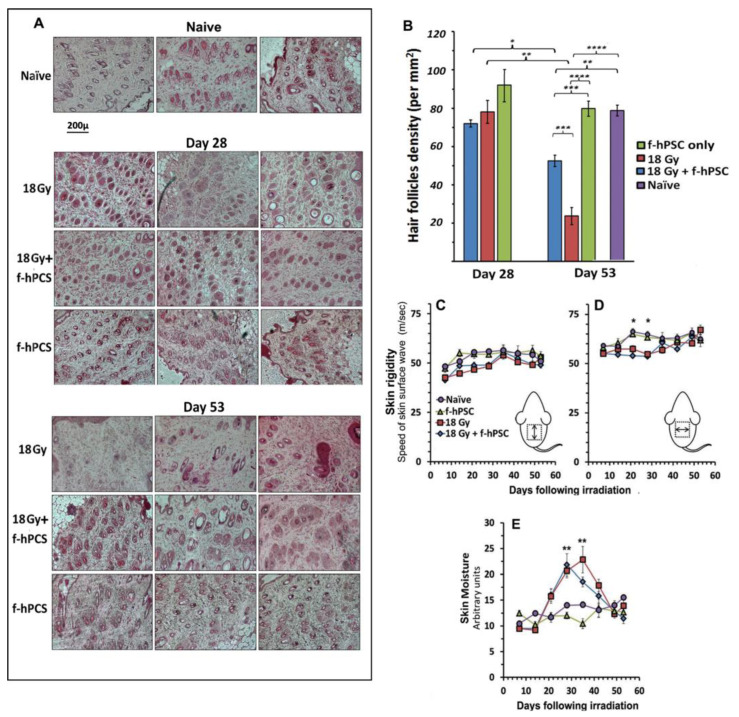
(**A**) Sections in the upper dermal layer parallel to the skin surface for morphometric histological quantitation of hair follicle density of the hair follicle density in the experiment of 18 Gy irradiation of the skin flap only. Half sacrificed by day 28 after irradiation and the rest on day 53 in representative mice from the different groups. All sections were photographed at the same magnification, as given as a bar under the panel of the naïve group. (**B**) A summary of the results of morphometry hair follicles density on the in the skin sections at the 2 time points tested. While on day 28 the density of the follicles was not significantly different from the controls, it dropped sharply by day 53 in the irradiated skin. Though still significant, this change was much less significant in the f-hPSC treated group. A very significant higher hair density was observed in the f-hPSC treated Vs non-treated irradiated mice. Directional follow-up measurements of skin rigidity by VESA showed some reduction of skin rigidity between weeks 3–5, which was non-significant for measurement in the longitudinal medial direction (**C**), but was significant in the lateral/medial direction (**D**). In parallel, skin surface moisture was elevated very significantly in all irradiated group relative to the two control groups +/− f-hPSC treatment (**E**). (* *p* < 0.05, ** *p* < 0.01, *** *p* < 0.005, **** *p* < 0.001).

**Figure 5 ijms-23-13321-f005:**
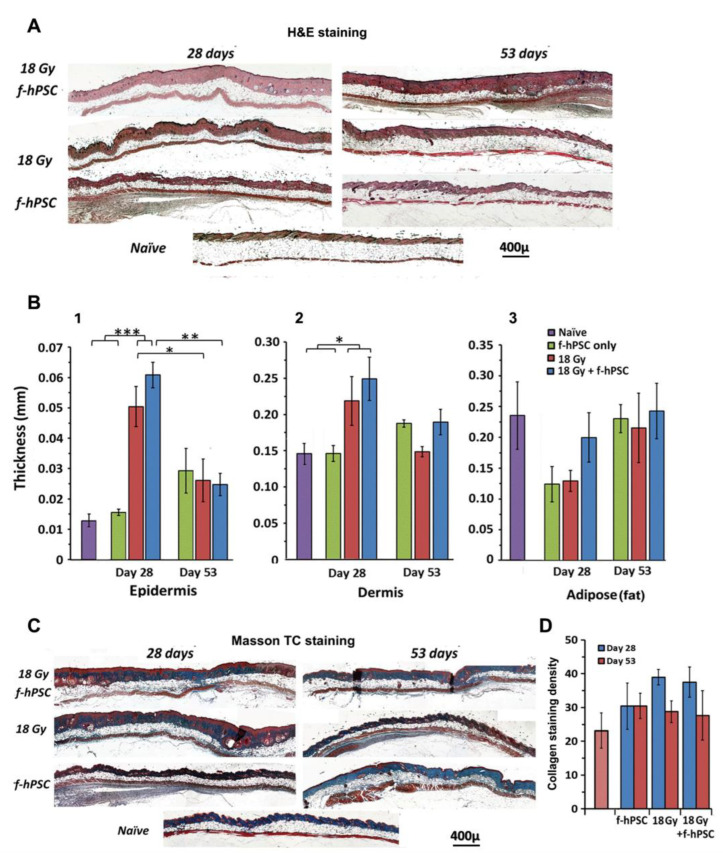
(**A**) Presentation of H&E staining of histological skin cross-sections of mice from main groups tested of 18 Gy irradiated +/− f-hPSC treatment and f-hPSC treated controls in non-irradiated mice. (**B**) Detailed morphometric analysis of the average thickness of the different skin layers Based on data from all the skin sections in the different tested groups, B1-epidermis, B2-Dermis, B3-adipose tissue. A significant increase of epidermal thickness in the irradiated groups at the early phase of day 28, which was reduced towards normal values by day 53. The slight non-significant reduction of adipose tissue thickness was apparent in an earlier phase of the follow-up. (**C**) Sections stained with Masson’s trichrome to detect collagen deposition and fibrosis. (**D**) Morphometric analysis based on blue pixel counts per dermal area for all the groups tested in the 2 time points. A non-significant apparent increase of collagen levels in the irradiated mice was evident in on day 28, which decreased towards normal values by the end of the termination of the study on day 53 (* *p* < 0.05, ** *p* < 0.01, *** *p* < 0.005).

## Data Availability

All the data on which this study is based are available upon request from the corresponding author at rafi@hadassah.org.il.

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
