# Peer review of "Alleviation of Severe Skin Insults Following High-Dose Irradiation with Isolated Human Fetal Placental Stromal Cells"

_ijms, 2022, doi:10.3390/ijms232113321_

Round 1
Reviewer 1 Report
Concerns:
1. Figure 3 was not included in submitted manuscript.
2. Please include rationale for having 2 f-hPSC treatments post irradiation, and for the timing of the 2 treatments.
3. The description of your results for Fig. 4 D&E should be presented more clearly to reflect that irradiation produced significant changes in lateral rigidity/skin surface moisture that f-hPSCs did not rescue.
4. The discussion should include thoughtful speculation on how f-hPSCs aid in post-irradiation effects. It was mentioned (line 100-101) that this is likely regulated through the secretome. Expansion of this idea in the discussion, along with the potential for further studies, would strengthen this work.
Minor concerns:
5. Please proofread for grammar, punctuation, spelling throughout
6. Define abbreviation for BM-MSCs on line 72 (currently defined on line 285).
7. Line 134 states injections were performed at 2 and 8 weeks post irradiation. This is meant to be 3 and 8 weeks?
8. Two statistical tests are mentioned in your methods. It should be clarified which test you are using in your figure legends.
9. Line 278: Reference is made to Fig 6B. It should be Fig. 5D.
10. Line 280-283: please separate this last sentence to clarify that this conclusion is based on all of your data (not the last figure).
11. Please justify the use of all female mice in your study, or address in the discussion the point that sex-dependent differences cannot be analyzed in the current study.
Figure comments:
12. Line 112: Direct reader to the relevant panel in Fig. 1
13. Describe Fig1. Panel D in text (are lack/expression of these markers consistent with f-hPSCs?); Include passage information for cells in panel D.
14. For all figure panels with graphs that include “Days after irradiation” on the x axis, it would be helpful to represent this in weeks since that is how you refer to time in your text. (Also spell check “irradiation” in Fig. 2A)
15. Be consistent with naming the control group. Is the “control” group in Fig. 2C the same as f-hPSCs? If not, please clarify.
16. Figures 2-5. It would be helpful to the reader if the experimental groups (f-hPSCs, 16/18 Gy, 16/18 Gy + f-hPSCs) were presented in the same order throughout.
17. Include Figure 3 upon resubmission.
18. Fig. 5: Panel A. It would be helpful to include a magnified example that labels skin layers analyzed (i.e., epidermis, dermis, adipose). Panel C/D. Your analysis of collagen should be restricted to the dermis. Images in panel C should be cropped and zoomed in to better appreciate collagen staining. Was your analysis in panel D confined to the dermis? Or full skin? The graph in panel D has 2 groups which are labeled 18 Gy. Please correct.
Reviewer 2 Report
Reviewer report for manuscript ijms-1929937
The manuscript "Alleviation of severe skin insults following high dose irradiation with isolated human fetal placental stromal cells" aimed to evaluate the role of cells isolated from the fetal placenta tissue (f-hPSC) on the healing process of the skin after exposure to a high dose of irradiation. Although the manuscript exhibits promising results, it should be carefully reviewed since it displays long periods, sentences hard to read, and phrases with duplicated words. The methodology should be better described (e.g., there is no mention in the section material and methods on how the cells were used). An essential part of the observed data is also missing (Figure 3). As a result, it is not possible to evaluate the correlation between these results and the respective drawn conclusions. The main highlight is the lack of discussion regarding the mechanistic role of f-hPSC on the skin's healing process. It is possible to find below suggestions for improvement.
Major comments
Although the manuscript exhibits promising results, a discussion about them is missing. Even though the study seems to be in its early stages, the authors are encouraged to not only report the observed data but also draw hypotheses to explain them. There are some questions below to help in this process:
(I) Why do subdermal f-hPSC injections seem to decrease the irradiation damage to the roots of the incisors? (Figure 2C)
(II) Why was the hair follicle density in the treated mice significantly higher than in irradiated controls and was not significantly different from the control? (lines 168-169, Fig 2E)
(III) Why, after 18Gy irradiation, the decrease in the hair follicle density was observed only in the late stage? (i.e., 53 days after the exposition, lines 220-221)
Section 2.3. - Figure 3 is missing. Which supposedly exhibits data from 18Gy irradiation assay. Therefore, it is not possible to evaluate the correlation between the obtained results and the drawn conclusions. Furthermore, regarding the decrease in the follicle density due to 18Gy irradiation, it was reported that "only moderate not significant decrease in hair follicles density in f-hPSC treated mice was monitored (Fig. 4B)." (lines 223-224). However, a comparative evaluation of figure 4B reveals the follicles density decreased from ~70 mm2 on day 28 to ~50 mm2 on day 53, i.e., decreased ~30%, and it was signed in Fig4B as significant, as well as in its note: "Though still significant, this change was much less significant in the f-hPSC treated group." (line 232). Therefore, please review this conflicting statement.
Minor comments
According to the "Instructions for Authors," a graphical abstract should be provided (https://www.mdpi.com/journal/ijms/instructions#preparation);
Abstract – In line 18 is reported "Both the head only and a back skin flap were irradiated by 6MeV electron beam". Please, declare that this irradiation was performed on mice.
Line 128 – It should be Fig. 2A instead of 1A
Figure 1D. – Please report in the Material and Methods the approach used to evaluate the surface markers phenotype of the cells and, to section results, references to support that f-hPSC cells display the referred surface markers.
Figure 2 - (A) Please add the standard deviation. (C) Add a footnote explaining the meaning of the letters R and B
Line 278 – It should be Fig 5D instead of 6B?
3. Discussion – Please, reserve this section to discuss the obtained results (most of them are in the section results). Moreover, move the first paragraph of this section to the introduction.
4. Material and Methods
The authors described the Isolation, expansion, and characterization of the f-hPSC. Nevertheless, there is no mention in the section materials and methods of how the cells were used. Please, reallocate the methodological description in section 2.2. to an appropriate one in section 4. Moreover, report the statistical significance of the observed moderate decrease in body weight (line 132)
4.3. Irradiation – It was reported that the protective shield allowed irradiation dose reduction of the unexposed areas (e.g., <0.4Gy line 372, < 1Gy line 377). Please, better describe the method applied to perform this evaluation.
4.5. Professional expert review of histology to quantify skin damages grading of skin condition – Please, better explain the referred grading scale. Is it based on an internal protocol or an international guideline? If the former, add to the supplemental material a section containing a table describing each stage of the grading scale to each evaluated parameter of the skin condition. If the latter, provide the appropriate reference.
4.6. Histology - Report the histological staining protocol. Additionally, describe how the thickness of the different skin layers was measured.
4.7. Statistical analyses - Provide the name and version of the software used to perform the statistical analyses.
Review the following sentences
Lines 53-59 – It seems that throughout the paragraph, the authors describe the advantages of in-vivo clonogenic assays. However, the periods are long and hard to understand. Please review it;
Lines 136-137 – "which was more significantly more pronounced in the untreated irradiated mice." "More" is duplicated;
Lines 141 – 144 "This resulted in an immediate sharp weight regain of the f-hPSC treated mice, so that by the termination of the experiment, on week 10, the mice in this group gained back their average weight, relative to controls and the f-hPSC treated mice (Fig. 2A)." It seems that "the f-hPSC treated mice" is duplicated.
Line 201 – "Almost a full recovery of the lesions was observed as observed at about 8 weeks"
Lines 214-215 – "the skin surface was done similarly to the procedure described for the preliminary experiment in the ?? (as previously shown in Fig. 2D)"
Lines 318-319 – "The current report further supports the feasibility of the use of the apparently complication-free f-hPSC treatments as a simple, and apparently complication free"
Round 2
Reviewer 1 Report
The manuscript has been improved.
My previous comment 16 [Figures 2-5. It would be helpful to the reader if the experimental groups (f-hPSCs, 16/18 Gy, 16/18 Gy + f-hPSCs) were presented in the same order throughout.] was not adequately addressed. Specifically, Fig. 2 D,E; Fig. 4 A,B; Fig. 5 A,B. The order in which images are presented in each figure should correspond in order to the quantified data in the following panel (eg.; if the images are presented from top to bottom 16 Gy, f-hPSCs, 16 Gy + f-hPSCs, they should be represented in the following panel's bar graph left to right 16 Gy, f-hPSCs, 16 Gy + f-hPSCs.)
I do recommend for this adjustment to be made prior to publication for ease of interpretation on the part of the reader.
Author Response
We thank the reviewer for mentioning this point which may have been misunderstood before. Al changes in the legend and orders on the panels within the graphs were taken care of. We appreciate the efforts of the reviewers which clearly improved the presentation of our resultsReviewer 2 Report
Reviewer report for manuscript ijms-1929937- v2
Although there are still a few suggestions for improvement, the manuscript improved significantly compared to its first version, mainly regarding the methodology and data presentation. Therefore, the reviewer recommends that, after performing the following suggestions for improvement, the manuscript would be suitable for publication,
Minor comments
Figures – Please, report the hair follicle density of the 28-day naïve group in fig 4B. Add a scale bar (as done in fig 2C) or inform the magnification to the optical microscope figures. Additionally, review the resolution of the figures as they look blurred, making it challenging to confirm statements like "[…] As clearly evident from these photographs" (lines 323);
In Materials and Methods (Section 4.5, line 710), the authors report that the data from VESA is exhibited in figures 3D & E. However, figure 3D is about the scoring of the skin damage, and 3E is missing. Afterward, in section 2.3 (lines 381-382), the authors report "[..] The follow-up with the VESA which was performed only in experiment 2 is presented in Fig. 4C&D." Nevertheless, this information was not reported in Materials and Methods (Section 4.5), making it possible to conclude that this analysis was performed in both experiments. Please review this conflicting information;
2.3. Effects of high dose irradiation restricted to a large skin flap with full protection of the rest of the body – It should be only Fig 3B in the following statement "the f-hPSC treatments were given in shorter time intervals from skin irradiation, as shown in Fig. 3A. &B." (lines 305-307). Moreover, the verb tense of Fig 3 caption should be in the past;
Figure 5B-3 – The adipose tissue of the non-irradiated group that received f-hPSC only seems to decrease in thickness 28 days after injection, exhibiting a pattern similar to the irradiated group (i.e., 18Gy). Please, explain this phenomenon;
Figure 5D - the far-most left bar is missing identification (is it the naïve group?)
In lines 411-413 it was reported "Morphometric evaluation was done by counting the blue stained pixels in the Mason's trichrome stained skin sections per the exact tissue area as translated from pixels to mm2 (Fig. 5B)" Please, describe in section 4.7 how the area as translated from pixels to mm2;
The second paragraph of the discussion section (lines 532-543), which describes the author's previous reports, does not seem to contribute significantly to the scope of the current discussion. Please, remove it.
4.3. Irradiation - The authors reported that the estimated exposure of the protected area to the irradiation was < 0.8Gy (lines 669-670). However, it is unclear how this value was calculated. Please, provide additional information and the appropriate reference to support this estimation;
4.4. Cells injection – While carrying out the large back skin flap irradiation assay, was the injection done in more than one site? Please, inform the site(s) of the injection.
4.7. Histology – As previously requested, please, describe how the thickness of the different skin layers was measured. Did the authors use software? How many measures were carried out? Was the measurement performed blindly? Additionally, follow the same instructions to inform how the hair follicle density was calculated;
4.8. Statistical analyses and data presentation – Please, provide the name and version of the software used to perform the statistical analyses, as previously requested, as well as in the "Instructions for Authors" (Research Manuscript Sections, subsection Materials and Methods: "[…] Give the name and version of any software used"
Please, review the following sentences
Lines (526-527) – "Therapies are proposed for the efficient protection of severe delayed effect of high dose irradiation are scarce". It is hard to read;
Lines (530-531) – "even when injected days or even weeks following exposure." Delete the second "even"
